# Socio-Ecological Factors That Influence Infant and Young Child Nutrition in Kiribati: A Biocultural Perspective

**DOI:** 10.3390/nu11061330

**Published:** 2019-06-13

**Authors:** Stephen R. Kodish, Kelsey Grey, Maryam Matean, Uma Palaniappan, Stanley Gwavuya, Caitlin Gomez, Tinai Iuta, Eretii Timeon, Martina Northrup-Lyons, Judy McLean, Wendy Erasmus

**Affiliations:** 1Departments of Nutritional Sciences and Biobehavioral Health, Pennsylvania State University, University Park, PA 16802, USA; 2Nourish Global Nutrition, Vancouver, BC V6H 4A7, Canada; kelsey.grey@nourishglobalnutrition.com (K.G.); maryam.matean@nourishglobalnutrition.com (M.M.); caitlin.gomez@nourishglobalnutrition.com (C.G.); nlyons.martina@gmail.com (M.N.-L.); judy.mclean@nourishglobalnutrition.com (J.M.); 3UNICEF Pacific, Suva, Fiji; upalaniappan@unicef.org (U.P.); sgwavuya@unicef.org (S.G.); tiuta@unicef.org (T.I.); werasmus@unicef.org (W.E.); 4Ministry of Health and Medical Services, Bikenibau, Republic of Kiribati; eretii1979@gmail.com

**Keywords:** formative research, Pacific Island Countries, qualitative methods, infant and young child nutrition

## Abstract

This study sought to elucidate the multi-level factors that influence behaviors underlying high childhood stunting and widespread micronutrient deficiencies in Kiribati. This two-phase formative research study had an emergent and iterative design using the socio-ecological model as the guiding theoretical framework. Phase 1 was exploratory while phase 2 was confirmatory. In phase 1, in-depth interviews, free lists, seasonal food availability calendar workshops, and household observations were conducted. In phase 2, focus group discussions, pile sorts, participatory workshops, and repeat observations of the same households were completed. Textual data were analyzed using NVivo software; ethnographic data were analyzed with Anthropac software for cultural domain analysis. We found a combination of interrelated structural, community, interpersonal, and individual-level factors contributing to the early child nutrition situation in Kiribati. Despite widespread knowledge of nutritious young child foods among community members, households make dietary decisions based not only on food availability and access, but also longstanding traditions and social norms. Diarrheal disease is the most salient young child illness, attributable to unsanitary environments and sub-optimal water, sanitation, and hygiene behaviors. This research underscores the importance of a multi-pronged approach to most effectively address the interrelated policy, community, interpersonal, and individual-level determinants of infant and young child nutrition in Kiribati.

## 1. Introduction

Maternal and child malnutrition, including both under- and overweight, are persistent global challenges that threaten child survival and development, contribute to a high disease burden, and damage the economic productivity of individuals and societies. Undernutrition, which encompasses acute and chronic undernutrition, as well as micronutrient deficiencies, is linked to more than one third of child deaths worldwide [1]. For those who do survive, chronic undernutrition in the critical period between conception and a child’s second birthday—the first 1000 days—can lead to irreversible impairment of physical and cognitive development, which reduces school performance and adult earning capacity, perpetuating the cycle of undernutrition and poverty [2]. Stunting, an indicator of chronic undernutrition, which results from undernutrition during the first 1,000 days, is recognized as one of the most substantial barriers to human development and affects approximately 156 million children under five years of age (U5) worldwide. This statistic includes about 30% of children from several Pacific Island Countries where stunting remains the foremost public health problem [3].

One of those countries is the Republic of Kiribati, a lower-middle income nation comprised of 32 atolls and coral islands with an estimated 110,000 people in 2015; it has a per capita GDP of <$2000 and one of the highest rates of U5 mortality in the Micronesia region (55 deaths/1000 live births) [4,5,6]. A high burden of child undernutrition, maternal overweight and obesity, and micronutrient deficiencies exists on Kiribati [7,8,9,10]. Children U5 years are disproportionately affected: 34% and 37% are stunted or anemic, respectively [3,9]. Approximately 80% of Kiribati women are overweight and 50% are obese [5], which is particularly important during pregnancy: maternal obesity may have negative consequences both on the foetus (e.g., large-for-gestational age infants, congenital malformations, stillbirth) and on the newborn, who is at higher risk of obesity, diabetes, and cardiovascular diseases, later in life [11]. Maternal and child diets are therefore two critical aspects of any nutrition situation.

In Kiribati, however, the dietary factors contributing to the sub-optimal nutrition situation of Kiribati are not fully clear. Previous health and nutrition surveys revealed relatively good breastfeeding practices: early initiation of breastfeeding is 80% [7], exclusively breastfed 0–5 months of age is 69%, with shorter mean duration in urban (3.5 months) than in rural islands (4.4 months) [12]. Complementary feeding practices are less optimal: just 31% of children aged 6–23 months have minimal acceptable diets, likely due to widespread food insecurity [7]. The proportion of young children fed ≥3–4 food groups in the past 24 h varies by urban (67.7%) and rural (48.8%) Kiribati, highlighting geographic differences in young child diets [7]. One well-documented factor is the high food prices and low household incomes: 50% of the average Kiribati household income is spent on food purchases, creating a situation whereby the most nutritious complementary foods are largely inaccessible [13,14].

The water, sanitation, and hygiene (WASH) environment is also an important immediate factor of early child nutrition in this context. Nearly 70% of rural Kiribati households have no access to basic sanitation facilities and 15% of urban households lack access to any type of toilet [15]. Less than a third of mothers report handwashing before eating (28%) and after cleaning a child’s feces (23%) [16]. Diarrhea accounts for over a tenth (13%) of under-five deaths in Kiribati, one of the highest in the Pacific Island countries [17]. Environmental enteropathy, a condition caused by chronic fecal exposure, is a contributing factor to young child growth faltering and undernutrition [18].

It is clear that diet and infection are key immediate determinants of infant and young child nutrition in Kiribati. However, there is very little understanding of the contributing underlying factors. Therefore, this formative study in Kiribati was designed to (1) assess determinants of food availability and accessibility across seasons; (2) describe the underlying social and behavioral determinants of child undernutrition comparing urban and rural settings; and (3) generate context-specific recommendations to inform tailored intervention strategies.

## 2. Materials and Methods

### 2.1. Study Setting

Teaoraereke, South Tarawa, of Central Gilbert Islands and Butaritari of the Northern Gilbert Islands, were selected as the urban and rural data collection sites in Kiribati, respectively. The official language of Kiribati is English but most people speak the native Micronesian language, Gilbertese (I-Kiribati). Roman Catholics (57.3%) and Protestants (Kiribati Uniting Church) (31.3%) represent the primary religious denominations [19].

#### 2.1.1. Teaoraereke (South Tarawa)

Urban data collection was conducted in Teaoraereke, South Tarawa, which has a population of 5105 [4]. Agricultural conditions are poor, as soil conditions are substandard and fresh water is scarce; there are also periodic droughts and a low annual rainfall of approximately 928.3 mm [20]. The food supply and water and sanitation systems are challenged by its high population density (3184 people per km^2^), contributing to limited land for agriculture [21]. Residents mainly obtain food by purchasing it, and increasing poverty is causing a reliance on cheap, imported foods such as refined rice, sugar, and flour, as well as foods high in fat, salt, and sugar [22], a trend reflective of changing dietary patterns in the Pacific region at large with negative population-level health effects [23].

#### 2.1.2. Butaritari

Butaritari, an island comprised of 11 villages, was selected for rural data collection. It has a population of 3224 and a density of 322 people per km^2^ [4]. Butaritari benefits from more favourable growing conditions than South Tarawa, with more arable land and approximately 1678.9 mm rainfall per year [20]. These favorable conditions allow households to grow coconut, breadfruit, giant taro (a tropical plant with starchy root vegetables), banana, and pandanus (a tropical tree with pineapple-like fruits), among other fruits and vegetables. However, food prices are generally higher than those in urban South Tarawa, contributing to food access challenges for typical households [22].

### 2.2. Study Design and Data Collection Methods

This formative study was designed to include participatory, mixed methods drawn from Focused Ethnographic Study procedures for informing nutrition interventions [24]. Qualitative and ethnographic data were collected from March until May 2018 over two iterative phases (Table 1) by 7 locally-hired data collectors who were selected based on their computer proficiency, language abilities, secondary school education level or higher, and previous experiences conducting health-related fieldwork in Kiribati.

#### 2.2.1. Exploratory Phase 1: Understanding Biocultural Influences on Infant and Young Child Nutrition

During phase 1, semi-structured interviews were conducted with male and female caregivers of children aged 6–23 months, community leaders (e.g., elected officials, religious leaders, traditional male leaders known as *unimwane*), health workers (e.g., nurses, community health volunteers, traditional healers), and senior-level health staff (e.g., district-level or national-level health staff from the Ministry of Health and Human Services) to elicit individual narratives illustrating health and nutrition perceptions and practices (Appendix A). Primary caregivers typically identified as mothers in this study, but also included fathers, grandparents, and other relatives to a lesser extent and depending on individual household dynamics. Participants were asked questions covering a range of topics related to food availability and security, maternal and child health, infant and young child feeding (IYCF) practices, and WASH. Interviews lasted 45–60 min each and were guided by unique semi-structured interview guides per participant type (Appendix A). Free lists were conducted among caregivers to understand the cognitive domains (i.e., local body of knowledge pertaining to a specific topic) related to young child foods and illnesses and specific to the Kiribati cultural context.

Seasonal Food Availability Calendar workshops were conducted to describe food availability across seasons. Workshop participants included local foods experts, including farmers, food vendors and consumers who were tasked with creating a calendar outlining all foods available by season in their community (i.e., urban or rural Kiribati). For each food, participants indicated (1) no availability, (2) low availability, (3) medium availability, or (4) high availability, before coming to a consensus over accuracy.

Direct observations were used to describe the intra-household factors that influence dietary patterns of infants and young children. Both mealtime and full-day (i.e., from a child’s first until last meal for 10–12 h) observations were conducted to reveal food preparation methods as well as feeding and hygiene behaviours. Using a semi-structured form, both behavioral events and time were recorded as they occurred, as well as at least every 10 min during continuous behaviours (e.g., cutting vegetables) (Appendix A). These observations were repeated among the same households and children in phase 2 to reduce reactivity.

#### 2.2.2. Confirmatory Phase 2: Identifying and Organizing Multi-Level Factors of Infant and Young Child Nutrition

Phase 2 built on the findings and themes generated from Phase 1. Focus group discussions among caregivers of infants and young children 6–23 months were used to identify social norms and to clarify interview findings. The focus groups were conducted separately by gender, without the presence of community leaders, due to cultural norms that may have hindered participants from speaking. Participatory community workshops were facilitated among diverse community members who brainstormed and voted upon the top barriers and suggested strategies to address nutrition in this setting. These workshops have been successfully used elsewhere to actively engage people and develop bottom-up intervention strategies [25,26].

Pile sorts used salient (*S* > 0.30) free list items (i.e., those top free listed terms scoring 0.30 or higher according to Smith’s salience index rank (*S*), which is calculated by both frequency and order of item mention across participants) to assess local food and illness classification systems [27]. Caregivers sorted stacks of cards, with words written on them and representing either salient childhood illnesses or young child foods, based on their perceived similarities and differences. Finally, walkthrough observations were conducted using structured checklists to describe environmental conditions influencing community nutrition and focused on water and sanitation structures, accessible food stands, kiosks, and markets, as well as provisions for animal husbandry [28].

### 2.3. Sampling

The Social Ecological Model (SEM) framed our study design and guided data collection including the purposive sampling procedures that guided this formative work [29,30]. The SEM is a useful and comprehensive framework from which to conceptualize the multi-level factors influencing nutrition-related behaviors, offering an advantage over other health behavior and food choice models which tend to emphasize individual-level determinants. For this work, potential participant types were delineated based on their level of influence according to the SEM (Table 2).

Second, within each participant type, socio-demographic criteria deemed important for ensuring a range of nutrition perspectives were chosen. Third, local health workers (e.g., nurse aids working at community health clinics) assisted in recruitment of eligible participants by making initial household visits based on their familiarity with local communities. Fourth, sample sizes were based on the estimated number of people appropriate for each method to either reach ‘data saturation’ for textual data analysis (interviews, focus groups, observations), ensure validity for cultural domain analysis (free lists, pile sorts), or allow for comfortable facilitation of participatory methods (participatory community workshops, seasonal food availability workshops) (Table 3).

### 2.4. Data Analysis

#### 2.4.1. Textual Data Analysis: Interviews, Focus Groups, Direct Observations

Interviews and focus groups were conducted in the Kiribati language and were audio recorded using digital recorders. Then, two locally hired team members transcribed and translated the audio files verbatim into English transcripts. Drawing from Grounded Theory, broad themes and sub-themes pertinent to the study aims were first identified across transcripts [31,32]. These themes were then incorporated into an analytic codebook, which reflected both the interview guide content and any newly identified thematic areas (Appendix A). Within NVivo 12 Software [33], the codebook guided line-by-line identification and systematic labelling of pertinent themes across transcripts. All coded text was then stratified by participant or method type, extracted based on each research question, and interpreted by triangulating with relevant findings from other employed methods. Descriptive field notes taken by the research team during fieldwork were also used to contextualize findings and interpretations.

#### 2.4.2. Cultural Domain Analysis: Free Lists and Pile Sorts

Free listed items for each participant were entered into Anthropac 4.98 [34]. A salience statistic was calculated for each item based on their rank order [27,35]. The relative salient young child foods and illnesses were interpreted in consideration of interview findings for more complete ethnographic perspectives. Pile sort items were chosen if they had a salience cut-off of >0.30 or were deemed important enough for addressing the study aims. For instance, breastmilk was not identified to be a salient ‘young child food’ during free lists but was included in pile sorts due to its nutritional importance.

Pile sort data were entered into Anthropac 4.98, which calculated aggregate proximities and generated item-by-item matrices for items with cells indicating the proportion of times two items appeared in the same pile across participants. Multi-dimensional scaling was used to analyze those aggregate proximity matrices in the software [35]. The goodness-of-fit, or stress, was also calculated for each matrix and is indicative of the strain remaining when the items in a cultural domain have been fitted into two dimensions. Stress values range from 0 (worst fit) to 1 (best fit). Two-dimensional visual maps were generated, with classifications labelled based on field notes.

#### 2.4.3. Participatory Workshop Analysis: Seasonal Food Availability and Participatory Community Workshops

Seasonal food availability workshop data were entered into a customized template. Symbols used to represent the relative availability of foods across seasons were translated into numerical values following standardized analytic procedures [36,37]. The food items listed by workshop participants were grouped according to nationally determined food groups (i.e., ‘body-building’,’ protective’, and ‘energy’ foods) to create a color-coded calendar depicting food availability across the year by season. The numerical data from other participatory community workshops were analyzed using simple arithmetic to tally and then rank the top-voted barriers and intervention strategies.

#### 2.4.4. Ethical Approval

The study was conducted in accordance with the Helsinki Declaration. Prior to any data collection, oral informed consent of all participants was obtained by the data collectors. The Kiribati Ministry of Health and Medical Services granted ethics approval prior to all research activities.

## 3. Results

This formative study drew findings resulting from 56 in-depth interviews, 94 pile sorts, 84 free lists, 11 focus groups, 10 participatory workshops, and 20 household observations completed across urban and rural sites (Table 3). The average age of interview participants was 39.4 years (21–71 years range), primarily female (64.3%), and split between urban (46.4%) and rural (53.6%) sites. The large majority of participating health workers were female (90.9%) whereas most community leaders were male (83.3%). The average age of young children being observed was 14.5 months.

### 3.1. Food Availability and Accessibility

Despite widespread knowledge of the most nutritious and healthful locally-available foods in urban and rural Kiribati, data suggest that household decisions around food procurement and subsequent dietary patterns are primarily based on what is available and accessible across seasons.

#### 3.1.1. Availability

In both urban and rural areas, fresh fish and shellfish and processed foods such as rice, bread, noodles, and tinned meats are highly available throughout the year. However, the availability of fresh local fruits and vegetables (e.g., breadfruit, pandanus, papaya) is greater in rural Butaritari due to more favourable growing conditions, including higher rain fall and greater availability of growing space (Table 4 and Table 5).

Barriers to improving food availability through homestead food production, rather than relying on the delocalized food system, include firstly the lack of available arable land in urban Kiribati.
“It’s very difficult to plant any vegetables here because of no space so we just plant pawpaw and breadfruit trees. Also, it’s very difficult to plant cabbage because of no space. If we move to this side it’s blocked and to the other side it’s the same thing. There’s also no space to grow pumpkins. Our source of fruit is the breadfruit and pawpaw, nothing else.”Male caregiver interview, South Tarawa [urban]

Also, insufficient supplies needed for agriculture and fishing in the rural outer islands was cited as a second barrier.
“For growing crops, they (community members) need tools. Shovels, spades, those are the things they need. If they don’t have them then it would prevent them from working in their gardens. If you want to start working on it but you can’t because you don’t have any tools.”Community leader interview, South Tarawa [urban]

Participants explained that a primary barrier to home food production was the lack of individual motivation to produce one’s own food. The relative convenience of buying and preparing processed foods (i.e., those with added sugars, salts, and/or fats for flavor enhancement and palatability, as well as additives and use of manufacturing technologies to allow for increased durability (shelf life) for long-distance shipping of food items [38]) outweighs the effort needed for regularly consuming local foods (e.g., breadfruit or taro) and was found to be an important determinant of present-day adult and child dietary patterns, representing a marked shift from those of the past.

#### 3.1.2. Financial Accessibility

Household food access in Kiribati is further constrained by the high cost of the limited available fresh fruits and vegetables, which are reportedly more affordable when they are home grown. However, due to very limited local food production, imported, processed foods such as rice, tinned fish, corned beef, and milk are dietary staples despite their high prices relative to local household incomes in markets.

### 3.2. Maternal Nutrition

Dietary intake among pregnant and lactating women in Kiribati is largely determined by what foods are available and accessible to them across seasons. As a result, maternal diets in this setting are disproportionately high in non-nutritious, energy-dense food items, including those high in starch (e.g., imported rice), saturated fat (e.g., tinned meats such as SPAM), and added sugar (e.g., sodas).

Food choices are also guided by specific cultural rules that reflect a medical belief system within which both food prescriptions and food proscriptions (i.e., taboos) influence dietary decisions.

#### 3.2.1. Food Prescriptions

Pregnant women are encouraged by health workers and family members to eat a “balanced diet” consisting of three food groups (a protein, a vegetable, a starch) per meal.
“When I was pregnant my diet was always balanced because I eat young noni (common fruit-bearing tree native to the Pacific region) shoots, papaya, pumpkin… I eat them because I want my baby in my womb to be healthy and have a balanced diet.”Female caregiver interview, Butaritari [rural]

Health workers perceptions and advice influence what mothers and child eat: they urge pregnant and lactating women to drink toddy (a syrupy drink made of coconut tree sap) as well as to eat coconut, fish, and shark meat for increased breastmilk production.
“Those kinds of foods that I mention, like drinking toddy and eating coconut, those are the most common foods that I eat only when I’m breastfeeding.”Female caregiver interview, Butaritari [rural]

Women’s mothers and husbands primarily support pregnant women during pregnancy by assisting with household tasks such as cooking and laundry as well as childcare.

#### 3.2.2. Food Proscriptions

Culturally, pregnant women are expected to avoid certain culturally proscribed foods that are said to impact the healthy development and physical appearance of a child (Table 6).

These expectations exist not only among pregnant women and other community members, but also among health workers and senior-level health staff, indicating social norms.
“Don’t ever eat octopus. It’s a source of protein they have but women don’t eat those kind of things because the baby will be bald…and even with some fish with bigger eyes, don’t eat those kinds of fish…it gives the same feature to your children but it’s [perception] changing now.”Senior-level health staff interview, Ministry of Health and Medical Services

Data suggest that these cultural rules were more influential in the past yet still persistent today, particularly in the rural outer islands where these food taboos were mentioned more frequently among older aged participants including grandparents and senior health workers. Health-seeking behaviors are governed by cultural rules within the medical belief system of Kiribati that recognizes both clinical and traditional medicine approaches.

Today, health workers operate also from a biomedical perspective, encouraging women to limit their intake of salty and oily foods to prevent high blood pressure and other health complications during pregnancy. Health worker misconceptions around the causes and treatment of anemia do remain though. For instance, some health workers explained that “excess salt intake” and “eating bar soap” during pregnancy were causes of anemia.

### 3.3. Infant and Young Child Feeding

#### 3.3.1. Intra-Household Dynamics

Kiribati households, including young children aged 6–23 months, typically ate three meals a day. Infants and young children were favored during mealtimes in terms of sequence of eating: the youngest children were fed before older children and before adults. Observations revealed similar feeding practices, in terms of dietary quality or quantity, based on child gender.

#### 3.3.2. Gender Norms

Traditional gender roles persist in both rural, and to a lesser extent, urban Kiribati: culturally, the primary female caregiver is expected to feed infants and young children. Grandmothers are important influencers of mothers, expected to provide dietary advice and child care support as needed. Fathers are responsible for household food procurement, but are not expected to feed or care for children to the same extent as maternal household members. Male caregivers’ lack of motivation and (in)ability to consistently procure nutritious foods were factors attributed to inadequate dietary intake among other household members.

Social activities outside the home, such as kava drinking and bingo playing, which are popular among Kiribati adults, were found to be competing demands by diverting caregiver time and household income away from otherwise positive IYCF practices.
“Like playing bingo and drinking kava. These are the two that they spend most of their time. Mothers can sit playing bingo from lunch time until dark and they forget their own responsibility for their children. They could spend money like $20 on kava rather than spend money on their children’s food.”Female caregiver focus group, South Tarawa [urban]
These social activities were similarly prevalent in both urban and rural communities.

### 3.4. Breastfeeding

#### 3.4.1. Early Initiation of Breastfeeding

Data support previous survey findings that breastfeeding practices are generally good and align with WHO recommendations. Participants in urban and rural Kiribati explained that the early initiation of breastfeeding is an important practice due to the benefits of colostrum for child health; this practice is encouraged by both traditional and professional health workers.

#### 3.4.2. Exclusive Breastfeeding

While professional health workers also counsel caregivers to exclusively breastfeed until six months, salient barriers to this practice were identified (1) the early introduction of complementary foods due to perceived inadequate breastmilk; (2) the perception that the introduction of complementary foods at three months of age can protect children from illness; (3) the practice of giving traditional medicine in the first days of life. The most commonly described traditional medicine is a liquid extract of pandanus root and pandanus leaf shoots given for the treatment of “buru,” an illness that is perceived to be with newborns at birth.
“Just after giving birth to my child they gave the child the local medicine for ‘buru’ [child sickness]. We hid it from the nurse when we gave it to the child. Sometimes it was prepared at home and they bring it to the hospital and we give the child that liquid to drink but without the nurse knowing it. After that there’s no other liquid that I gave my child.”Caregiver interview, South Tarawa [urban]
“Only my medicine for the ‘buru’ [child sickness]. This is the first liquid that I give my grandchildren when they are first born. If not given, it could lead to death to the baby. So it is most important that that medicine is given.”Traditional healer interview, Butaritari [rural]

This illness is thought to cause the child to appear blue (as in cyanosis) and cry a lot. (4) Competing demands, with women returning to work soon after giving birth, was identified as a salient barrier.

Data suggest that while grandparents are important household members to support with feeding and care practices, they are also key household influencers most likely to introduce complementary foods early, as well as to use traditional medicines to address infant and young child health.

#### 3.4.3. Continued Breastfeeding until Two Years or Beyond

Breastfeeding is perceived to be essential for child health and continued breastfeeding until two years or beyond is commonly reported by urban and rural caregivers, as well as by health workers. However, there is a persistent perception that breastfeeding while pregnant will cause diarrhea for the breastfeeding child. It is also perceived that breastmilk flavor will negatively change if feeding during pregnancy. Therefore, breastfeeding is a proscribed behavior for pregnant women in Kiribati.

### 3.5. Complementary Feeding

Interview and observation data revealed that most children aged 6–11 months are fed meals of soft, mashed foods, which were prepared specifically for them, in addition to water, breastmilk, and other liquids. Older children aged 12–23 months typically ate the same foods as older children and adults, usually rice and fish. In both urban and rural Kiribati, the most salient young child foods were those locally available except for rice, which was most salient in urban South Tarawa and sixth most salient in rural Butaritari (Table 7 and Table 8).

The unstructured pile sort for the salient ‘young child foods’ in South Tarawa revealed four clusters, with a stress of 0.132 (Eigenvalue: 13.677; EigenRatio: 4.844), revealing local categorizations of foods in this cultural context (Figure 1).

The unstructured pile sort for young child foods also revealed four clusters, and a stress of 0.085 (Eigenvalue: 20.589; EigenRatio: 3.168), illustrating local food classification systems. (Figure 2).

Children U2 years were rarely observed eating snacks between meals, but sugary drink consumption (e.g., toddy and sugar water) was common; they were also commonly fed non-nutrient dense foods such as “swimming rice” (a watery mixture of rice and water or milk), pumpkin mashed with water, or high-sugar drinks. Like maternal diets, children’s diets are influenced greatly by what is available/accessible, as well as the relative convenience of procuring and preparing processed foods compared to locally available foods.

#### 3.5.1. Introduction of Complementary Foods

Among participants, there was differential introduction of complementary foods. Those caregivers who reported the timely introduction at 6 months had received a positive interpersonal influence from health workers. Those caregivers who introduced complementary foods earlier (e.g., at 3 months) were often influenced by older community members or grandparents who indicated that those foods were needed for “keeping a body strong” and “protecting against illness”. Much less common were caregivers who introduced foods only at 8–12 months of age. Interview data indicate that these caregivers delayed introducing food due to previous experiences with early food introduction and bouts of childhood illness.

#### 3.5.2. Style of Feeding

Direct observations and interviews revealed a variety of different feeding styles, with the most commonly observed style being “laissez-faire,” particularly among children older than 12 months. Responsive feeding practices (i.e., feeding infants directly, assisting older children when they feed themselves, being sensitive to children’s hunger and satiety cues, encouraging children to eat, and offering different foods if children refuse to eat) were inconsistently observed. Observational data indicate that caregivers rarely displayed any of the characteristic feeding behaviors considered ‘responsive’ with little variation across participants. During child illness, feeding patterns did vary by household: some caregivers fed children starchy foods to stop diarrhea and others were fed only liquids.

### 3.6. Water, Sanitation, Hygiene

#### 3.6.1. Access to Safe Drinking Water

Access to rainwater tanks for collecting and storing water is the most salient and second-most salient challenge to safe water access in urban and rural Kiribati, respectively (Table 9 and Table 10).

The top-voted barriers to safe water access mentioned in community workshops were also discussed across interviews and focus groups.

#### 3.6.2. Perception of Illness Risk

The source of water influences perceptions of water safety and illness risk. Participants explained that well water is more commonly boiled than rainwater, which is perceived to be less contaminated. Community leaders emphasized that in reality very little boiling occurs for any type of water due to the inconvenience despite higher risk perception among caregivers toward water-borne illness than nutrition-related illness.
“The preparation of food is not well carried out and also because the majority of the people here are very lazy in boiling water…and they do not have rain water, it is the norm for them to drink sometimes dirty water that they saved in their container and sometimes the container is not clean and they just drink it. In that way they get diarrhea.”Community leader interview, South Tarawa [urban]

Diarrhea was found to be the most salient childhood illness in both urban and rural Kiribati. However, its perceived seriousness for young child health differs greatly among community members. Most household observations and walkthroughs revealed infants and young children who, while playing and learning to walk, were sharing environs with unfenced pigs and chickens. Participants explained that the accumulation of animal faeces in a fenced area creates unpleasant odors and prevents their easy disposal. Land availability is very limited, particularly in urban South Tarawa; thus, communities are characterized by spaces shared by both developing children and domestic animals.

#### 3.6.3. Hand Washing Behaviors

Hand washing with soap at critical times (i.e., after toilet use and before handling food) hadhistorically not been practiced. Its adoption was inconsistently observed across households. This is also due to challenges associated with limited indoor plumbing. Without this infrastructure, handwashing requires someone to access rain or well water before using two separate, outdoor basins for washing. The same basins are used for dishwashing, with one containing soapy water and the other non-soapy water for rinsing.

Interview data suggest that individual understanding of the relationship between soap, infection, and health was variable by interview participant and based on previous education and/or exposure to WASH interventions. Three primary strategies (e.g., food safety, water treatment, personal hygiene) were cited by participants for diarrheal prevention and reflect high community-level awareness stemming from previous WASH sensitization efforts. In comparison, prevention strategies for nutrition-related illnesses, such as stunting, are much less widespread and known.

#### 3.6.4. Defecation Practices

Open defecation is a socially accepted norm across Kiribati despite high community-level awareness that toilet use could prevent disease. The inadequate supply of toilets for the population, coupled with the poor maintenance of those in existence, were identified as key barriers to their habitual use.

## 4. Discussion

A combination of structural, social, and behavioral factors at multiple levels of influence synergistically contribute to the sub-optimal nutrition situation in the early years of life in Kiribati. By taking a biocultural perspective to this formative research, multi-level opportunities were identified for improving the health and nutrition situation, specifically focused on the first 1000 days of life.

At the policy level, the delocalized food system of Kiribati relies heavily on imported, non-perishable food items that are largely non-nutritious, yet highly accessible. Sustainable nutrition solutions necessitate strong political will focused on multi-sectoral approaches. Infrastructure investment to facilitate ease of local food trade from outer islands to South Tarawa may improve availability and access for urban households. Policies to encourage healthier local food choices should be championed, such as the 2014 excise taxes, which imposed a levy on unhealthy foods, including sugar-sweetened beverages [22,39]. Fresh food subsidies and product labelling may address undernutrition among young children as well as maternal overweight later in life [40,41]. The provision of treated piped water and the construction of structurally sound wells should constitute continual policy-related WASH efforts backed by evidence-informed frameworks [42]. Such structures are the building blocks of WASH- and nutrition-friendly environments [43].

At the community level, longstanding social norms around food and illness, such as reliance on traditional medicine in early life and food proscriptions during pregnancy, are factors of sub-optimal diets. Trying to change underlying cultural rules, at least in the short term, may be in vain. However, context- and age-specific counseling interventions in other traditional settings where longstanding social norms also strongly influence IYCF [44,45] have improved breastfeeding practices, highlighting the potential for also doing so in Kiribati. Further, harnessing the strength of local governance structures and social networks to improve health and nutrition-related behaviors has particular potential in Kiribati considering the vast distances among outer islands and strongly interdependent communities in this context. Community participation in the planning, implementation, and evaluation of community-based interventions is an effective strategy for implementing successful public health programs [46]. Village welfare groups are community self-help bodies selected by community leaders in the maneaba system of Kiribati governance, which is an important community influence. These groups are headed chairpersons who are selected by the community to serve as role models. Each group has an assigned nurse to support the delivery of health-related programming and thus are an existing structure to utilize for health promotion efforts. These community groups are well-positioned to act as agents of health behavior change in Kiribati, yet require re-prioritization in national health plans which have de-emphasized their importance in recent years [47].

At the interpersonal level, Kiribati culture is driven by interdependent construals of the self, a communal society whereby social influence and community standing are paramount drivers of behavior [48]. This social dynamic is an opportunity for positive health and nutrition intervention. Community gatherings both in maneabas and in other venues should be utilized for mobilization, whereby participatory strategies are employed to involved groups of community members who want to drive change. While interpersonal communication is the preferred channel of caregivers to receive health information, the combination of tailored interpersonal communication and mass media, which is becoming increasingly popular, has been effective in other resource-constrained settings [49]. A tailored combination of communication channels should include culturally-appropriate SBCC strategies to reach and diffuse among audience segments and Kiribati social networks.

Church leaders and traditional community leaders, such as unimwane (male elders), are strong influencers who are well positioned for health promotion. The church has been used successfully elsewhere for positive nutrition-related intervention, serving as a platform with trusted influencers to reach audiences with messages that resonate and are linked to core cultural values of religions [50]. Not only church leaders, but also community members can provide personal testimonials related to positive health and nutrition experiences [51]. Because the church is a place where social gatherings occur on a weekly basis, it should also be used for having regular cooking demonstrations and healthy food fairs for instance. These events should aim to include both primary caregivers as well as other key influencers such as fathers/husbands, grandparents, and other household members (e.g., adolescents) whom we found to be important for IYCF in both urban and rural Kiribati. Using comedic songs, dance groups, and theatre programming to promote nutrition and WASH practices during gatherings in the maneaba may reach a range of different audience segments across Kiribati communities where traditional education models have been found to be less effective [52]. After all, comedic dances and sketches are already central to maneaba gatherings, where Kiribati culture values over-the-top performances.

Individual perceptions toward dietary and WASH practices also varied across participants. However, such practices do not occur in a vacuum: individual differences were influenced by a combination of underlying and basic factors such as limited food availability and access. Therefore, individual-level behavior change should only be expected once other multi-level factors are addressed [53]. For instance, we found high caregiver knowledge of foods that are non-nutritious yet difficulty avoiding them due to their affordability, convenience, and positive flavor profiles, which are developed early in child development [54]. Positive individual level perceptions were also observed: breastmilk was thought to be nutritious for children by most participants, thanks to previous campaigns by professional health workers who are well respected in both the urban and rural islands.

Individually, however, intervention design can focus on increasing risk perception toward nutrition-related illness. Perceiving a health threat is the most obvious prerequisite for the motivation to change risky behaviors [55]. This study found very low perceived risk of nutrition-related illnesses and thus it is difficult to expect easy dietary changes during intervention. Because people respond adaptively to health risk feedback, SBCC efforts in Kiribati should therefore seek to raise awareness about benefits of illness avoidance, as well as the consequences of not doing so. A combination of advocacy and social mobilization, as well as individual-level counselling from health workers and interpersonal nudges from respected elders, may slowly change risk perception toward nutrition-related illnesses [56]. Tailored messages should not be so clinical in nature, but instead employ persuasive communication strategies (e.g., appealing to positive effects) as well as appeal to Kiribati core cultural values, such as communalism [57,58,59]. Of course, SBCC is more than just a tool to raise individual perception of risk toward illness but a tailored set of approaches for improving many individual-level, psychosocial determinants of nutrition practices including subjective norms, attitudes, and self-efficacy, just to name a few. Drawing from lessons learned from previously successful or unsuccessful SBCC campaigns in Kiribati around other health and nutrition topics will be important for informing effective intervention design.

Individual handwashing and boiling of drinking water are not yet habitual for most people in Kiribati. That is, individuals report that they oftentimes just forget to do these everyday actions and that they require cognitive planning/thinking. Sustained health behaviors need to be consistent to be effective [60]. Consider the habit of wearing a seat belt in many settings, for instance, which is less determined by risk perception as it is an engrained habit in our behavioral DNA and performed largely without forethought [61]. To get there, behavioral prompts, which can change behaviours at the point of decision making, can be effective strategies in the short term by serving as reminders at key times [62]. Behavioural prompts that are well suited for Kiribati include print media reminders (i.e., locally-developed stickers or small posters) placed in the kitchens of households or next to household wells and basins, as well as radio jingles made by community members explaining the importance of boiling water to prevent illness. Handwashing and boiling water are two exemplar behaviors that require habituation for sustenance, requiring not only reminders but also the requisite structures (e.g., indoor plumbing) to allow for convenient and easy behavioral follow through [63].

### Strengths and Limitations

During this study, data were triangulated by using multiple data sources, different data collectors, multiple analysts, and mixed data collection methods. Doing so helped to ensure data credibility and gave us confidence in our findings, as triangulation is a key strategy for ensuring data credibility in qualitative research [64]. Also, this study was designed with two iterative phases, a luxury of qualitative research whereby emergent themes relevant to the study objectives could be identified in one phase and clarified or confirmed in the next [65]. Finally, a strength of this work was the continued presence of the research team members during the entirety of fieldwork. Doing so allowed for participant observation of communities and households during that time, an important aspect of conducting ethnographic research to fully contextualize and understand local health and nutrition practices [66].

This study did have some limitations. Possible reactivity to the presence of the data collectors observing practices was noted during fieldwork, which is a common threat to validity in direct observation of health behaviors [67]. That is, participants may have changed their behaviours while being observed by the research team. Repeated observations of the same households were designed to reduce reactivity, a strategy employed in other work aimed to understand nutrition-related behaviors [68]. All reactive behaviours were also recorded and accounted for during analysis. That is, after coding observational data and synthesizing findings, we referred to field notes about reactivity to ensure conclusions we drew were based on those behaviors we felt were natural and triangulated by other data sources. Second, our research team introduced themselves as government staff members. It is possible that respondents’ urge to give socially desirable answers with the perception that they would receive development assistance or other incentives affected the findings [69]. However, through multiple forms of data triangulation we are confident that final interpretations are sound. Finally, the data in this study was only collected during one season of the year although efforts were made to collect data that reflected seasonal differences (e.g., seasonal food availability workshops).

## 5. Conclusions

Multi-level behavioural determinants of optimal health and nutrition in the early years of life (from conception to the child’s second birthday) were identified through this formative research in urban and rural Kiribati. While myriad barriers exist at individual, interpersonal, community, and policy levels, numerous facilitating factors to good child health and nutrition were also elucidated through this work. Kiribati is a unique country context whose communities would benefit greatly from culturally-appropriate and tailored social behaviour change intervention strategies.

## Figures and Tables

**Figure 1 nutrients-11-01330-f001:**
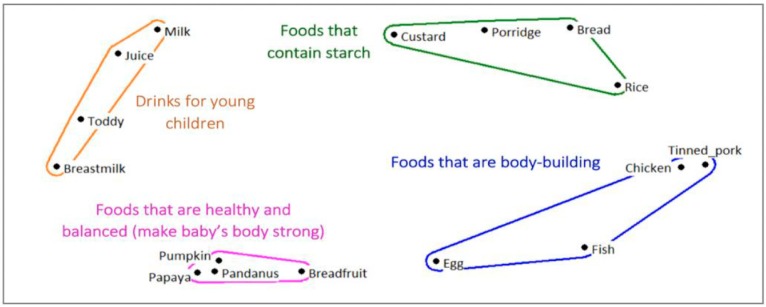
Multi-dimensional scaling map of pile sorting ‘young child foods’ in South Tarawa [urban].

**Figure 2 nutrients-11-01330-f002:**
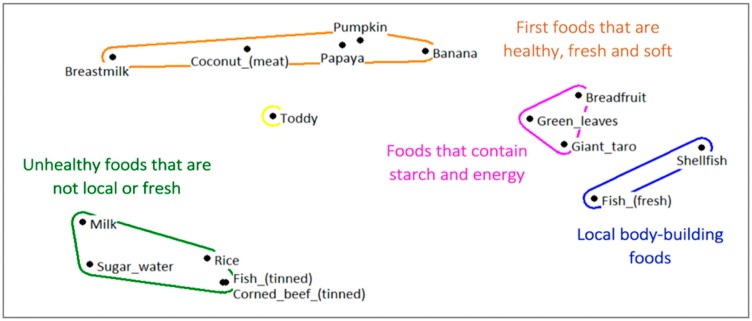
Multi-dimensional scaling map of pile sorting ‘young child foods’ in South Tarawa [urban].

**Table 1 nutrients-11-01330-t001:** Formative study design with data collection methods by study phase.

Exploratory Phase 1	Analysis and synthesis	Confirmatory Phase 2
In-depth interviews	Phase 1 data analysis	Focus group discussions
Free lists *	Phase 2 instrument development using Phase 1 findings	Pile sorts **
Seasonal food availability calendar workshops		Participatory community workshops
Household observations I		Household observations II

* Free listing is a cognitive anthropology method to elicit salient items of a cultural domain (e.g., ‘young child foods’ and ‘illnesses’); ** Pile sorting is a sorting activity using top free list terms to reveal local classification systems of a cultural domain (e.g., local food groupings).

**Table 2 nutrients-11-01330-t002:** Initial sampling framework with participant types by level of influence.

Level of Influence	Participant Types
Policy	Senior-level health staff (e.g., Ministry of Health and Human Services)
Organizational	Professional and traditional health workers
Community	Community leaders
Interpersonal	Fathers, grandparents
Individual	Primary caregivers (typically female mothers)

**Table 3 nutrients-11-01330-t003:** Final sample sizes and participant types by data collection method.

Data Collection Method	Sample Size (*n*)		
	Urban	Rural	Total
In-depth interviews	26	30	56
Female caregiver	11	10	21
Male caregiver	5	5	10
Health worker	5	6	11
Senior health staff	-	-	2
Community leader	5	7	12
Pile sorts	41	53	94
Free lists	39	45	84
Focus group discussions	5	6	11
Female	3	3	6
Male	2	3	5
Community workshops	4	4	8
Household observations	14	6	20
Seasonal food availability workshops	1	1	2

Sampling procedures followed local customs and norms specific to each data collection site.

**Table 4 nutrients-11-01330-t004:** Seasonal food availability calendar for Teaoraereke village in South Tarawa [urban].

	Jan	Feb	Mar	Apr	May	Jun	Jul	Aug	Sept	Oct	Nov	Dec
Seasons	Dry season	Hot season (humid)	Storm Season	Rainy season
**Food**
**Energy Foods**
Rice												
Breadfruit *												
Bread products (doughnuts, buns)												
Noodles												
Cassava ^1^												
Weet-bix cereal												
Custard powder												
Mature coconut meat *												
Cooking oil												
**Body-building Foods**
Fish (fresh) *^,2^												
Mackerel (tinned)												
Pork (tinned)												
Pork (fresh) *												
Chicken ^3^												
Sausages (packaged)												
Clams *												
Eggs *^,4^												
Sea worms (dried or fresh) *												
Red meat (lamb or beef)												
Milk												
Bottled baby food ^5^												
**Protective Foods**
Island cabbage *												
Papaya *												
Pumpkin *												
Banana *												
Pandanus *												
**Sugary Drinks**
Coconut water *												
*Toddy **												
Milo (fortified malt drink mix)												
Orange juice												
*Toddy *syrup *^,6^												
**Other**
Sugar												
Salt												


 High availability 
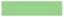
 Medium availability 

 Low availability 
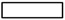
 No availability. * Locally produced foods. ^1^ Respondents indicated that cassava found in South Tarawa is imported from Fiji. ^2^ While respondents indicated that there are slight variations in the availability of tuna and flying fish, some type of fish is highly available at any given time in the year. ^3^ Respondents indicated that the chicken available in South Tarawa is imported and that it is not available in January or August as supplies run low after feast days in December (Christmas) and July (International Day of Kiribati). ^4^ Both imported and local eggs are available. The seasonal variation in eggs may be explained by the timing of shipments of imported eggs. ^5^ Bottled baby food is usually Heinz brand smooth custard which is a milk-based product. ^6^
*Toddy* syrup is a thick condensed coconut sap syrup. It is usually served diluted in water.

**Table 5 nutrients-11-01330-t005:** Seasonal food availability calendar for Butaritari island [rural].

	Jan	Feb	Mar	Apr	May	Jun	Jul	Aug	Sept	Oct	Nov	Dec
Seasons	Dry season	Hot season (humid)	Storm season	Rainy season
**Food**
**Energy Foods**
Rice												
Breadfruit *												
Mature coconut meat *												
Cassava *												
Giant taro *												
Flour												
Noodles												
Crackers												
Native fig *^,1^												
Sweet potato *												
Bread products												
Weet-bix cereal												
Young coconut meat *												
**Body-building Foods**
Fish *^,2^												
Shellfish *^,3^												
Crustaceans (crab, lobster) *												
Octopus *												
Eel *												
Mackerel (tinned)												
Chicken *												
Pork (fresh) *												
Dog meat *												
Tinned meat (all types)												
Turtle meat *												
Bottled baby food ^4^												
**Protective Foods**
Papaya *												
Pumpkin *												
Noni fruit *												
Taro leaf *												
Banana *												
Pandanus *												
Lime *												
Leafy green (nambere) *												
Island cabbage *												
Germinated coconut *												
Half-food (kaikere) *^,5^												
Eggplant *												
Cucumber *												
**Sugary Drinks**
*Toddy* *												
**Other**
Sugar												
Sugar cane *												


 High availability 
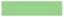
 Medium availability 

 Low availability 
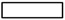
 No availability. * Locally produced foods. ^1^ Native fig is the fruit of ficus tintoria, the starchy immature fruits are pounded into a flour, preparation is labour intensive. ^2^ While respondents indicated that there are slight variations in the availability of tuna and flying fish, some type of fish is highly available at any given time in the year. ^3^ Respondents mentioned many types of shellfish, all of which are highly available throughout the year. ^4^ Bottled baby food is usually Heinz brand smooth custard which is a milk-based product. ^5^ Respondents mentioned a fruit that could not be identified by the data collectors, the literal translation of its name ‘kaikere’ is ‘half-food’. It is similar in appearance to mango with a pungent smell and bitter melon-like taste.

**Table 6 nutrients-11-01330-t006:** Food proscriptions during pregnancy and possible birth outcomes for infants.

Proscription during Pregnancy	Health Effect from Consumption during Pregnancy
Octopus	Child may be bald or have white spots on skin
Shark meat	Child may get angry easily or small like a shark
Blowfish	Child may be born without eyebrows or thinning hair
Nuonuo fish	Child may have gap teeth
Hot chilis	Child may be blind
Mon fish, turtle meat	Child may have big eyes
Pandanus lobes (cracked)	Child may have cleft palate
Lobster	Child may have crooked eyes

**Table 7 nutrients-11-01330-t007:** Salient young child foods from free lists among caregivers in South Tarawa [urban].

Overall Rank	Food Item (Kiribati)	Food Item (English)	Salience
1	Raiti	Rice	0.662
2	Ika (fresh; all varieties)	Fish (fresh; all varieties)	0.656
3	Mai, kabuibui	Breadfruit	0.578
4	Bwaukin	Pumpkin	0.524
5	Bwabwaia	Papaya	0.355
6	Bunimoa	Egg	0.243
7	Katitati	Custard	0.218
8	Taaman	Fish (tinned)	0.198
9	Moa	Chicken	0.184
10	Marai	Green coconut meat	0.184

**Table 8 nutrients-11-01330-t008:** Salient young child foods from free lists among caregivers in Butaritari [rural].

Overall Rank	Food Item (Kiribati)	Food Item (English)	Salience
1	Mai, kabuibui	Breadfruit	0.748
2	Bwaukin	Pumpkin	0.683
3	Marai, ben	Coconut meat	0.568
4	Bwabwaia	Papaya	0.554
5	Bwabwai	Giant taro	0.507
6	Raiti	Rice	0.462
7	Ika (fresh; all varieties)	Fish (fresh; all varieties)	0.428
8	Banana, green banana	Banana	0.367
9	Karawe	Toddy	0.255
10	Katitati	Custard	0.184

**Table 9 nutrients-11-01330-t009:** Top-voted barriers and solutions to clean water access in South Tarawa [urban].

# of Votes *	Top-Voted Barriers to Accessing Clean Water	Top-Voted Solutions to Accessing Clean Water
76	No rain water tank	Ask government to provide rain water tanks/funds for rain water tanks
48	No firewood/fuel to boil water	Educate communities on SODIS (solar disinfection) Ask government to lower the price of firewood
38	Poor quality well water (salty, contaminated by showers, toilets, mud, and animal feces due to inadequate well walls and covers	Build higher walls for wells (ask government to provide materials to do so)
22	No PUB (Public Utilities Board), water/pipes not working	Build own wells Report problems to the PUB PUB water to be distributed to each household, pipes to be functional
15	Septic tank contaminates water sources	Keep septic tank away from water source by 30 m Build compost toilet (as they do not require a septic tank)

* The symbol ‘#’ refers to ‘number of participant votes’ in table above.

**Table 10 nutrients-11-01330-t010:** Top-voted barriers and solutions to clean water access in Butaritari [rural].

# of Votes *	Top-Voted Barriers to Accessing Clean Water	Top-Voted Solutions to Accessing Clean Water
365	Poor quality well water (salty, contaminated by showers, toilets, mud, and animal feces due to inadequate well walls and covers	Build toilets and showers away from well Cover wells/build higher walls for wells Ask government to provide walls for wells
38	No rain water tank	Ask government to provide rain water tanks/funds for rain water tanks Desalination of sea water Assign security to watch over tanks to prevent damage
35	Not boiling water	Communication efforts to emphasize the importance of boiling water Educate communities on SODIS (solar disinfection) Have pots to boil water
10	Rain water used for kava drinking	Provide more water tanks Limit kava ** drinking
9	No cement for building walls around wells	Ask government to provide funds for cement

* The symbol ‘#’ refers to ‘number of participant votes’ in table above; ** Kava is a locally-made, popular alcohol beverage made from indigenous western Pacific island plants.

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
