# Peer review of "Socio-Ecological Factors That Influence Infant and Young Child Nutrition in Kiribati: A Biocultural Perspective"

_nutrients, 2019, doi:10.3390/nu11061330_

Round 1

Reviewer 1 Report

Thank you for the opportunity to review this article about multilevel factors that influence infant and young child feeding in Kiribati. This was a well designed study and well written manuscript. I enjoyed reading it. The authors present important findings that could influence policy and programs in Kiribati.

Overall, my comments are mostly seeking clarification. 

I would encourage the authors to consider including the interview guides and semi-structured observation form as supplementary materials. This would provide more details for readers, and would also be a valuable resource.

Line 55 Is the early initiation figure from the 2009 DHS and the EBF figure from Global Nutrition Report which is from UNICEF data? Splitting the references so they follow their respective figures would help clarify the sources for the reader. Also please re-word “exclusively breastfed until 6 months of age” as the UNICEF indicator is “Percentage of infants 0-5 months of age who were fed exclusively with breastmilk” so this includes 1 month old and 5 month old infants and does not give an accurate estimate of children who were EBF for 6 months. Also, consider removing recent as the DHS is from 2009.

Line 101: who conducted each of the data collection methods?

Line 102: Please clarify who the caregivers are (here and throughout the manuscript), from Table 2 it appears that the primary caregivers were women (did they all identify as mothers, if so please state that), and then fathers and grandmothers were also included. Clearly listing these participants would be helpful especially with the focus on different levels of influence, which includes interpersonal.

Line 110-112: How were the foods determined for the SFAC? Were they based on the free lists? Were they all foods or foods specific to children and pregnant women? Consider not using SFAC as an acronym since it is not widely used.

Line 126: how were community members identified and recruited for the workshops? Who facilitated these workshops?

Line 139: how were participants recruited?

For table 2, please include the data collection method and number of participants for each and distinguish between rural and urban, or create a new table

Line 149, were interviews and FGDs audio recorded transcribed and translated? What language were the interviews conducted in?

Line 186: Can you provide summaries of the demographic characteristics of different types of study participants?

Line 232: was this quote from an urban community leader, was this also mentioned by rural community members/leaders?

Line 254: were there differences in food taboos among women in rural and urban areas?

Line 290: were infants and young children favored over adults as well as older children?

Line 301-308: were these outside activities also reported in rural areas

Line 311: how were feeding styles determined? Did you use constructs from the IFSQ or some other measure?

Table 8 and 9: should the right column be labeled top-voted solutions (rather than barriers)

Line 459: consider these two articles that addressed prelacteal feeds//traditional medicinal oral remedies (in different contexts) but that were given for similar reasons – and saw changes in these practices. If they do not seem relevant to the context there is no need to include them, but in both there were social norms around traditional remedies, which were overcome in the short and longer term. Perhaps there are examples from contexts that are more similar.

Mbuya MN, Matare CR, Tavengwa NV, Chasekwa B, Ntozini R, Majo FD, Chigumira A, Chasokela CM, Prendergast AJ, Moulton LH, Stoltzfus RJ. Early initiation and exclusivity of breastfeeding in rural Zimbabwe: impact of a breastfeeding intervention delivered by village health workers. Current Developments in Nutrition. 2019 Feb 28;3(4):nzy092.

Matare CR, Craig HC, Martin SL, Kayanda RA, Chapleau GM, Kerr RB, Dearden KA, Nnally LP, Dickin KL. Barriers and Opportunities for Improved Exclusive Breast-Feeding Practices in Tanzania: Household Trials With Mothers and Fathers. Food and Nutrition Bulletin. 2019 May 8:0379572119841961.

Line 475: can you say more about the influence of fathers, grandmothers and other family members? Since they were included in data collection. How do they relate to interpersonal influence?

Line 499: there is a typo

Line 501: Could you say more about the findings related to responsive feeding or the lack thereof?

Line 503: what about the benefits of optimal infant and young child feeding rather than only focusing on threats? There are other benefits of IYCF beyond illness that could be promoted similar to other successful programs (like Alive & Thrive which was mentioned elsewhere).

Author Response

I would encourage the authors to consider including the interview guides and semi-structured observation form as supplementary materials. This would provide more details for readers, and would also be a valuable resource.

Authors: Good suggestion. We have included these materials as supplementary materials.

Line 55 Is the early initiation figure from the 2009 DHS and the EBF figure from Global Nutrition Report which is from UNICEF data? Splitting the references so they follow their respective figures would help clarify the sources for the reader.

Authors: We have split these references based on your recommendation and hope it is now clearer for readers.

Also please re-word “exclusively breastfed until 6 months of age” as the UNICEF indicator is “Percentage of infants 0-5 months of age who were fed exclusively with breastmilk” so this includes 1 month old and 5 month old infants and does not give an accurate estimate of children who were EBF for 6 months.

Authors: Thanks, this is always a bit tricky but we do agree that since we referred to children aged ‘6 – 23 months’ when referring to complementary feeding practices then we should be consistent and indicate ‘0 – 5 months’ when discussing EBF. We have made this change in this section of the manuscript.

Also, consider removing recent as the DHS is from 2009.

Authors: Good point – we have replaced the word ‘recent’ with ‘previous’ which we agree is more accurate.

Line 101: who conducted each of the data collection methods?

Authors: All data collection was conducted by a locally hired team of 7 data collectors who were recruited and hired based on specific proficiencies which we have now clarified in the manuscript under section 2.2.  

Line 102: Please clarify who the caregivers are (here and throughout the manuscript), from Table 2 it appears that the primary caregivers were women (did they all identify as mothers, if so please state that), and then fathers and grandmothers were also included. Clearly listing these participants would be helpful especially with the focus on different levels of influence, which includes interpersonal.

Authors: We recruited caregivers of different varieties, including mothers, fathers, grandparents and other relatives responsible for the care of a child 6-23 months. We have clarified this in the manuscript and removed the reference to “female” under Primary Caregiver in Table 2 as we agree this term was limiting when in fact we sought to find different types of caregivers and not all households in this study had a primary caregiver who identified as female. Thank you for pointing this out.

Line 110-112: How were the foods determined for the SFAC? Were they based on the free lists? Were they all foods or foods specific to children and pregnant women?

Authors: All foods listed in the SFAC were identified by community members during the workshop as any food available in their community throughout the year, separate from the free listing activity. These community members included key informants who, together, would be able to provide a comprehensive and accurate picture of the food availability across seasons. They included agricultural experts such as farmers, as well as food vendors and consumers. We have added this clarification to the manuscript.

Consider not using SFAC as an acronym since it is not widely used.

Author: Yes, we agree that fewer acronyms is often better and therefore we have removed ‘SFAC’ from this manuscript as it had only been used 4 or 5 times throughout the paper.

Line 126: how were community members identified and recruited for the workshops? Who facilitated these workshops?

Authors: We have briefly clarified participant recruitment in section 2.3 Sampling, a step-wise process which reflects the purposive sampling procedures used for all data collection methods including community workshops. The workshops were also facilitated by our 7 locally-hired data collection team members; the word ‘moderator’ may have made this point unclear so we have removed this word as we see how it may have wrongly implied the workshops were led by moderators who were different from the 7 data collectors, but this was not the case.

Line 139: how were participants recruited?

Authors: We have briefly clarified participant recruitment in step 3 of section 2.3 Sampling section.

For table 2, please include the data collection method and number of participants for each and distinguish between rural and urban, or create a new table

Authors: We found that addressing your suggestion is best achieved through the addition of a new table – a new Table 3 – to include the sample sizes by urban/rural site by each data collection method. Where appropriate, we have indicated what types of participants were involved by type of method. Combining this information with that of Table 2 because a bit too messy and did not help the overall manuscript. We hope you find this Table 3 to be a good compromise.

Line 149, were interviews and FGDs audio recorded transcribed and translated? What language were the interviews conducted in?

Authors: All interviews were conducted in the local language (Kiribati) and audio recorded using digital recorders. The audio files were then transcribed and translated verbatim into English. We clarified these points in the beginning of section 2.4.1 – thanks for this good suggestion.

Line 186: Can you provide summaries of the demographic characteristics of different types of study participants?

Authors: We have added a short introductory paragraph to Section 3. Results which summarizes some of the available demographic characteristics collected by method of this study. Combined with the newly created Table 3 outlining more detailed sample sizes, participants, and methods, we believe that the description of the study sample is overall improved in this revision. Thank you for these helpful suggestions. 

Line 232: was this quote from an urban community leader, was this also mentioned by rural community members/leaders?

Authors: This quote is an exemplar and from an urban community leader; however, the same sentiment was reported by participants who resided in the rural area as well. In this manuscript we present quotes that we feel best represent the underlying theme or sub-theme but try to point out in text when findings differ by type of participant or geography, etc. For purposes of presenting these findings, and for brevity, findings pertain to both urban and rural areas unless otherwise specified. Quotes were selected based on their representativeness and clarity.

Line 254: were there differences in food taboos among women in rural and urban areas?

Authors: We did expect to see some differences in taboos by rural and urban area but our data do not indicate so. We hypothesize that had we been able to collect data from the farthest, most rural islands then some of our findings may have revealed additional food prescriptions/proscriptions and other culturally-bound dietary practices not yet influenced by delocalization of the food system. We did find some differences in mention of food taboo by participant age, however, with older caregiver/grandmothers discussing taboos more frequently than participants who were younger.

Line 290: were infants and young children favored over adults as well as older children?

Authors: Yes, as the caregiver would feed infants and young children before herself. We have edited this Section 3.3.1 intra-household dynamics to clarify this point as well as to explicate the type of favoritism that we observed (sequence of eating) in households.

Line 301-308: were these outside activities also reported in rural areas

Authors: Yes, these activities were reported in both urban and rural areas of Kiribati and we have added a sentence to make this point explicit at the end of Section 3.3.2 gender norms.

Line 311: how were feeding styles determined? Did you use constructs from the IFSQ or some other measure?

Authors: We did use a validated measure such as the IFSQ to determine and report the feeding styles observed. We used our detailed field notes from the household observations, triangulated with textual data from observations where caregivers provided narratives of feeding episodes with young children, to draw conclusions representing general feeding styles. We agree that for future work focused specifically on feeding styles as primary study aims a validated measure such as the IFSQ would be informative. For purposes of this particular study, which had a large range of research aims and questions around IYCF at large, we are confident that the data drawn from various methods could be used for making general conclusions surrounding feeding styles as reported in this manuscript without overstating findings.

Table 8 and 9: should the right column be labeled top-voted solutions (rather than barriers)

Authors: Yes! Thank you for catching this mistake, we have made the correction in the manuscript.  

Line 459: consider these two articles that addressed prelacteal feeds//traditional medicinal oral remedies (in different contexts) but that were given for similar reasons – and saw changes in these practices. If they do not seem relevant to the context there is no need to include them, but in both there were social norms around traditional remedies, which were overcome in the short and longer term. Perhaps there are examples from contexts that are more similar.

Mbuya MN, Matare CR, Tavengwa NV, Chasekwa B, Ntozini R, Majo FD, Chigumira A, Chasokela CM, Prendergast AJ, Moulton LH, Stoltzfus RJ. Early initiation and exclusivity of breastfeeding in rural Zimbabwe: impact of a breastfeeding intervention delivered by village health workers. Current Developments in Nutrition. 2019 Feb 28;3(4):nzy092.

Matare CR, Craig HC, Martin SL, Kayanda RA, Chapleau GM, Kerr RB, Dearden KA, Nnally LP, Dickin KL. Barriers and Opportunities for Improved Exclusive Breast-Feeding Practices in Tanzania: Household Trials With Mothers and Fathers. Food and Nutrition Bulletin. 2019 May 8:0379572119841961.

Authors: Thank you very much for these suggestions. We have reviewed these articles and believe referencing them in the discussion section when discussing possible intervention models is quite valuable. We now refer to them in the third body paragraph of the Discussion section when explaining that social norms may be difficult to change but context- and age-specific tailored intervention may still have positive effects on breastfeeding practices in similar traditional communities. Thank you!

Line 475: can you say more about the influence of fathers, grandmothers and other family members? Since they were included in data collection. How do they relate to interpersonal influence?

Authors: This is a good point. We have added a 1 – 2 sentences to emphasize the potential benefit of including such influencers in interpersonal approaches on page 16 within the Discussion. We had alluded to doing so but had not discussed such influencers explicitly in the previous draft and therefore we agree with your suggestion and believe this addition helps to enhance the discussion section. Thank you.

Line 499: there is a typo

Authors: Thank you for catching this typo, we have made the correction in the manuscript.

Line 501: Could you say more about the findings related to responsive feeding or the lack thereof?

Authors: We have edited Section 3.5.2 style of feeding to make findings related to feeding style a bit clearer for readers, although admittedly the section is still briefer than it could be had the entire manuscript focused on understanding feeding styles as a primary study outcome. We see feeding style as one piece of a larger IYCF story and hope that this revised section, coupled with the other IYCF-related findings, is now informative enough to paint a comprehensive IYCN picture in Kiribati for readers.

Line 503: what about the benefits of optimal infant and young child feeding rather than only focusing on threats? There are other benefits of IYCF beyond illness that could be promoted similar to other successful programs (like Alive & Thrive which was mentioned elsewhere).

Authors: Good point. We have added a sentence at the end of this Discussion paragraph focused on risk perception to highlight the possibility for SBCC to also influence other psychosocial factors at the individual level, such as self-efficacy, etc.

Reviewer 2 Report

Review

Manuscript title: Socio-ecological factors that influence early child nutrition in Kiribati: a biocultural perspective.

Thank you for giving me the opportunity to review this interesting and well-written paper.

This paper sought to identify, through qualitative mixed methods approaches, drivers of early child nutrition in Kiribati. This study contributes to the body of knowledge of factors influencing poor nutrition in infants in low- and middle-income countries and provides an in-depth insight of the mechanisms through which factors at all levels of the socio-ecological model drive poor nutrition. Key target points for interventions are also highlighted and discussed to improve the nutrition situation in this context.

Please find below some minor comments to consider to strengthen the manuscript:

Introduction

-        Lines 46-47: Please state the level of income of Kiribati (a lower-middle income country).

-        Please introduce the socio-ecological framework in this section so it then links clearly with the choice of adopting this framework as a guide for data collection/analysis and interpretation of findings. Alternatively, this could be placed in the methods section but a brief discussion of the usefulness of socio-ecological models to understand issues related to nutrition and health is required in the manuscript.

-        Please add some information around rural/urban differences in the prevalence of stunting/exclusive breastfeeding and complementary feeding practices if available so as to justify the focus on both urban and rural areas which is not clear from the introduction section. It would then link nicely with the study setting description.

-        Line 52 you state that maternal obesity may have negative consequences both on the foetus and later in life. Can you please explain how maternal obesity link to fetal growth and potentially undernutrition at/around birth and how this relates to the child’s nutritional status later in life. Please expand and add supporting references. It is important as the focus from your title/abstract and introduction is on “early child nutrition” and more specifically on undernutrition (stunting and micronutrient deficiencies) and so the reader needs to understand clearly why you are mentioning figures around maternal obesity and how this directly relates to the child’s nutrition. Is one of the hypothesis that women during pregnancy/lactation consume products that are energy dense but nutrient poor and therefore the fetus/young infants may lack key micronutrient for growth?

-        Throughout the manuscript, you use different terminologies to describe your outcome (e.g. nutrition situation; nutrition; health and nutrition; food, nutrition and illness, dietary patterns). It would be good to use the same terminology throughout the manuscript for consistency and not to confuse the reader.

-        As per my comment above, please replace nutrition situation on lines 70-73 by “child undernutrition” or “poor nutrition in young infants” so the objective of the study is clearer.

-        If one key focus of your study was to identify rural/urban differences (which is obvious from the way you presented the results), I would suggest to mention this in the objectives of the study at the end of the introduction.

Material and methods

-        Lines 86-88: you make an interesting comment on availability of cheap, imported foods that are high in fat, salt and sugar. Please add a sentence to say how these types of diets can potentially be linked to stunting/micronutrient deficiencies and add supporting references.

-        Lines 93-94: define taro and pandanus. I wonder if some of the local foods mentioned in the paper could be defined somewhere in the paper (either when mentioned in the text or as footnotes). This is important for readers not familiar with dietary practices in this setting.

-        Table 1: Add footnotes to define free lists and pile sorts (the table needs to be standalone).

-        Line 101, title 2.2.1, be specific and add “in infants” in the title.

-        Line 103: list examples of community leaders, health workers and senior-level health staff.

-        Line 107: can the interview guide be added as an appendix?

-        Lines 108-109: This sentence needs to be made clearer by defining some of the concepts (e.g. free lists; cognitive domains)

-        Line 112: Please specify how this list of food was compiled. Have you used previous dietary surveys to find out the types of foods consumed in these areas or any other national data?

-        Line 121: In the title, you say “nutrition and health” but in the title 2.2.1 you say “food, nutrition and illness”. Please standardise.

-        Line 130: Please define salient free list items and S>0.30 for a reader not familiar with this methodology.

-        Line 132: give examples of young child foods

-        Line 139-140: If you decide to not introduce the SEM in the introduction, then please justify here why you chose the SEM over other models available, the advantage of using this etc.

-        Line 141: “on” is missing between based and their.

-        Line 153: Please consider adding the codebook for analysis as supplementary material

Results

-        General comment in this section: please specify sample size for each activity in this section. In the methods you mention that the number of people interviewed was appropriate for each method but we would expect to see final sample sizes for each participatory method/interviews in this section.

-        Line 192: You mention processed foods such as rice, bread, noodles, tinned meats. Please add a reference to support your definition of processed foods. Clearly state what processed refers to.

-        Title 3.1.2. Please consider replacing title by financial accessibility

-        For section 3.2. to stay in, you have to make a clear link between maternal nutrition/maternal consumption of EDNP (energy dense nutrient poor) foods and early infant nutrition. This relates to my earlier comment in the introduction.

-        Line 256: define young noni shoots

-        Line 286: what does drinking soap mean?

-        Line 314: remove “the” before “both traditional and professional health workers”

-        Line 321: define pandanus rood and pandanus leaf shoots

-        Line 370: define toddy

-        Line 370-371: how do you define non-nutrient dense foods?

-        Tables 8/9: the titles on columns 2 are the same than in columns 1. I assume these should say “solutions”. Can you also specify in the title the people who voted for these and who suggested solutions. This is mentioned in the methods but it would be good to have this information from the tables. Also add a footnote to define “kava”.

-        Lines 423: suggest removing “and is not yet habitual at the population level” as this is not emerging from the study findings

-        Lines 428-429: Please consider rephrasing the sentence as not clear

Discussion

-        Line 544: You mention all reactive behaviours were also recorded and accounted for during analysis. Can you say how and what was done?

Author Response

Thank you for giving me the opportunity to review this interesting and well-written paper.This paper sought to identify, through qualitative mixed methods approaches, drivers of early child nutrition in Kiribati. This study contributes to the body of knowledge of factors influencing poor nutrition in infants in low- and middle-income countries and provides an in-depth insight of the mechanisms through which factors at all levels of the socio-ecological model drive poor nutrition. Key target points for interventions are also highlighted and discussed to improve the nutrition situation in this context.

Please find below some minor comments to consider to strengthen the manuscript:

Introduction

-        Lines 46-47: Please state the level of income of Kiribati (a lower-middle income country).

Authors: Good suggestion – we have made this income level explicit in paragraph 2 of the introduction.

-        Please introduce the socio-ecological framework in this section so it then links clearly with the choice of adopting this framework as a guide for data collection/analysis and interpretation of findings. Alternatively, this could be placed in the methods section but a brief discussion of the usefulness of socio-ecological models to understand issues related to nutrition and health is required in the manuscript.

Authors: We have taken your suggestion about adding clarification to the Methods section where we introduce SEM instead of doing so here in the introduction. While we see the benefit of further emphasizing the SEM, we carefully considered the manuscript length – already a bit long – in our decision to briefly describe our choice of SEM in the methods over doing so in the introduction.

-        Please add some information around rural/urban differences in the prevalence of stunting/exclusive breastfeeding and complementary feeding practices if available so as to justify the focus on both urban and rural areas which is not clear from the introduction section. It would then link nicely with the study setting description.

Authors: We have added a couple of sentences to the Introduction section paragraph #3 highlighting differences in complementary feeding practices/dietary practices by urban/rural Kiribati, based on 2009 survey data. It is the only source we could find with such detailed information and one reason this work was carried out. We agree that this addition helps the introduction section so thank you.

-        Line 52 you state that maternal obesity may have negative consequences both on the foetus and later in life. Can you please explain how maternal obesity link to fetal growth and potentially undernutrition at/around birth and how this relates to the child’s nutritional status later in life. Please expand and add supporting references. It is important as the focus from your title/abstract and introduction is on “early child nutrition” and more specifically on undernutrition (stunting and micronutrient deficiencies) and so the reader needs to understand clearly why you are mentioning figures around maternal obesity and how this directly relates to the child’s nutrition. Is one of the hypothesis that women during pregnancy/lactation consume products that are energy dense but nutrient poor and therefore the fetus/young infants may lack key micronutrient for growth?

Authors: We have added some clarification with respect to your point at the end of paragraph 2 of the Introduction section. This clarification has also been supported by a very useful citation from Aviram, Hod, & Yogev from the Int’l Journal of Gynecology and Obstetrics entitled, “Maternal obesity: Implications for pregnancy outcome and long-term risks –a link to maternal nutrition”. Thank you for this important suggestion which we believe has improved the beginning of this manuscript quality.

-        Throughout the manuscript, you use different terminologies to describe your outcome (e.g. nutrition situation; nutrition; health and nutrition; food, nutrition and illness, dietary patterns). It would be good to use the same terminology throughout the manuscript for consistency and not to confuse the reader.

Authors: We have given the full manuscript a re-read and edited various terms based on your suggestion looking specifically for consistency of terminology as well as precision when it comes to diction. Thank you for this important suggestion to improve the manuscript.

-        As per my comment above, please replace nutrition situation on lines 70-73 by “child undernutrition” or “poor nutrition in young infants” so the objective of the study is clearer.

Authors: We have made this replacement and agree that consistent (and precise) terminology is important.

-        If one key focus of your study was to identify rural/urban differences (which is obvious from the way you presented the results), I would suggest to mention this in the objectives of the study at the end of the introduction.

 Authors: We have edited the second research aim to make this focus explicit.

Material and methods

-        Lines 86-88: you make an interesting comment on availability of cheap, imported foods that are high in fat, salt and sugar. Please add a sentence to say how these types of diets can potentially be linked to stunting/micronutrient deficiencies and add supporting references.

Authors: We have added a phrase to further contextualize this descriptive sentence, as well as a citation to support it. Thank you for making this recommendation.

-        Lines 93-94: define taro and pandanus. I wonder if some of the local foods mentioned in the paper could be defined somewhere in the paper (either when mentioned in the text or as footnotes). This is important for readers not familiar with dietary practices in this setting.

Authors: In this Butaritari sub-section 2.1.2 we have defined ‘taro’ and ‘pandanus’ in parentheses to provide just enough description for the reader to envision the type of food mentioned. We have also reviewed the rest of the manuscript and added brief explanations of local (emic) terminology where needed.

-        Table 1: Add footnotes to define free lists and pile sorts (the table needs to be standalone).

Authors: We have added two footnotes to this Table 1 defining free lists and pile sorts

-        Line 101, title 2.2.1, be specific and add “in infants” in the title.

Authors: We have edited the title of 2.2.1 and 2.2.2 to indicate ‘infant and young child nutrition’ which is now also consistent with the title of the manuscript and better aligned with the purpose of this work. Thank you for making this suggestion for enhancing the title precision.

-        Line 103: list examples of community leaders, health workers and senior-level health staff.

Authors: We have added examples of these types of participants for clarification in parentheses within this section. 

-        Line 107: can the interview guide be added as an appendix?

Authors: Interview guides have now been added as supplementary materials.

-        Lines 108-109: This sentence needs to be made clearer by defining some of the concepts (e.g. free lists; cognitive domains)

Authors: We have defined free lists in Table 1 so will refrain from doing so again here in this paragraph. However, we like your suggestion to define ‘cognitive domain’ and have done so here per your suggestion here in sub-section 2.2.1.

-        Line 112: Please specify how this list of food was compiled. Have you used previous dietary surveys to find out the types of foods consumed in these areas or any other national data?

Authors: This was a participatory workshop whereby community members brainstormed and collaborated to develop this list of foods. No national data or other secondary literature were needed for this activity, which was completed with the help of a research team moderator but done so primarily by the participants themselves. We have edited this paragraph to make this process clearer.

-        Line 121: In the title, you say “nutrition and health” but in the title 2.2.1 you say “food, nutrition and illness”. Please standardise.

Authors: We have edited the manuscript to be more consistent with our terminology – thank you for pointing out this inconsistency.

-        Line 130: Please define salient free list items and S>0.30 for a reader not familiar with this methodology.

Authors: We have added a clarification here, as well as a citation to support this statistic. It is also explained a bit more, and supported with citations, in the cultural data domain analysis sub-section 2.4.2.

-        Line 132: give examples of young child foods

Authors: The young child foods are actually one of our findings as they result directly from the free listing exercise and thus we would prefer to present them firstly in the results and not here in the Methods section.

-        Line 139-140: If you decide to not introduce the SEM in the introduction, then please justify here why you chose the SEM over other models available, the advantage of using this etc.

Authors: In light of word count, we have added one sentence to explain why we believed the SEM was most appropriate for framing this formative nutrition research. We agree that even this brief description is helpful for readers and hope the reviewer agrees.

-        Line 141: “on” is missing between based and their.

Authors: Thank you – we have added this preposition to the sentence.

-        Line 153: Please consider adding the codebook for analysis as supplementary material

Authors: The analytic codebook has now been added as supplementary material.

Results

-        General comment in this section: please specify sample size for each activity in this section. In the methods you mention that the number of people interviewed was appropriate for each method but we would expect to see final sample sizes for each participatory method/interviews in this section.

Authors: We have added Table 3 which includes sample sizes by type of participant, method, and urban/rural divide. Thank you for this suggestion to improve the quality of the manuscript.

-        Line 192: You mention processed foods such as rice, bread, noodles, tinned meats. Please add a reference to support your definition of processed foods. Clearly state what processed refers to.

Authors: We have used parentheses to define what we mean by processed foods in this manuscript and use a 2014 citation from Obesity Reviews to support this description, which can be found now on the bottom of page 8 of results.

-        Title 3.1.2. Please consider replacing title by financial accessibility

Authors: Good suggestion – we have made this change for specificity.

-        For section 3.2. to stay in, you have to make a clear link between maternal nutrition/maternal consumption of EDNP (energy dense nutrient poor) foods and early infant nutrition. This relates to my earlier comment in the introduction.

Authors: Thanks to your earlier comment and resulting edit, we believe this point is now clear. Also, we are confident that the readership interested to read this paper will be well educated about the important of maternal nutrition for fetal and child health outcomes; therefore, we opt to leave this section as it is for now.

-        Line 256: define young noni shoots
Authors: Noni is a common fruit-bearing tree in Kiribati indigenous to the Pacific region; its young shoots are consumed. We have added a parenthetical explanation of this local food.

-        Line 286: what does drinking soap mean?
Authors: Digesting soap was a phenomenon that was reported in Kiribati and was accepted that pregnant women would get cravings and eat soap (i.e. biting into a bar of soap); the word ‘drinking’ has been replaced with ‘eating’ as it was a diction error in text. Thanks for pointing this out.

-        Line 314: remove “the” before “both traditional and professional health workers”

Authors: Thank you for reading so carefully – we have deleted this word from section 3.4.1.

-        Line 321: define pandanus rood and pandanus leaf shoots

Authors: Pandanus is a common fruit tree in Kiribati and different parts of the plant are consumed (e.g. the fruit, the root, and leaves). We have defined this local word previously in the manuscript thanks to your earlier suggestion and thus will leave this one as it is.

-        Line 370: define toddy

Authors: This local term is described once in the food availability seasonal calendar footnote as well as in sub-section 3.21. and therefore we do not believe another definition is needed for toddy.

-        Line 370-371: how do you define non-nutrient dense foods?

Authors: We are referring to those foods that are lacking the vitamins and minerals needed for healthy growth and development and believe the readership of this paper will be informed enough to interpret this phrase appropriately in the broader context of maternal and child nutrition without further description.

-        Tables 8/9: the titles on columns 2 are the same than in columns 1. I assume these should say “solutions”. Can you also specify in the title the people who voted for these and who suggested solutions. This is mentioned in the methods but it would be good to have this information from the tables. Also add a footnote to define “kava”.

Authors: Thank you for this suggestion – we have made these edits thanks for your close attention to detail. We have also defined ‘kava’ in a footnote.

-        Lines 423: suggest removing “and is not yet habitual at the population level” as this is not emerging from the study findings

Authors: We agree this statement was more of an assumption than a fact/finding and we have agreed to remove it.

-        Lines 428-429: Please consider rephrasing the sentence as not clear

Authors: We have rephrased this sentence to be clearer.

Discussion

-        Line 544: You mention all reactive behaviours were also recorded and accounted for during analysis. Can you say how and what was done?

Authors: We have clarified this point in this paragraph and agree that such additional description was needed for the readers.